# Stoichiometric Variation in Soil Carbon, Nitrogen, and Phosphorus Following Cropland Conversion to Forest in Southwest China

Mengzhen Lu [1,2,3], Kunping Liu [1,2,3], Lijin Zhang [4], Fuping Zeng [1,2,3], Tongqing Song [1,2,3], Wanxia Peng [1,2,3] and Hu Du [1,2,3,*]

1 Laboratory of Agro-Ecological Processes in Subtropical Region, Institute of Subtropical Agriculture, Chinese Academy of Sciences, Changsha 410125, China; lumengzhen19@mails.ucas.ac.cn (M.L.); lkp18125@163.com (K.L.); fpzeng@isa.ac.cn (F.Z.); songtongq@isa.ac.cn (T.S.); wxpeng@isa.ac.cn (W.P.)
2 Guangxi Key Laboratory of Karst Ecological Processes and Services, Huanjiang Observation and Research Station for Karst Ecosystems, Hechi 547100, China
3 Guangxi Industrial Technology Research Institute for Karst Rocky Desertification Control, Nanning 530200, China
4 Research Center on Ecological Science, Jiangxi Agricultural University, Nanchang 330045, China; zhanglijin2022@163.com
* Correspondence: hudu@isa.ac.cn; Tel.: +86-731-84619713

**Abstract:** Soil organic carbon (SOC), nitrogen (N), and phosphorus (P) are three essential soil nutrients for plant growth, and their stoichiometric ratios are already important indices of elemental balance and the soil fertility status in soil ecosystems. The evolution mechanism of the SOC, Total Nitrogen (TN), Total Phosphorus (TP), and stoichiometry following the "conversion of cropland to forest program" (CCFP) in southwest China is not yet clear. Seven different CCFP restoration models, including *Zenia insignis* (RD), *Toona sinensis* (XC), *Castanea mollissima* (BL), *Citrus reticulate* (GJ), *Zenia insignis* and Guimu-1 elephant grass (RG), Guimu-1 elephant grass (GM), and abandoned cropland (LH), were chosen to explore changes in the concentration and stoichiometry of the SOC, TN, and TP, and their recovery times, at a depth of 0–100 cm. The results indicate that the SOC and TN concentrations in different restoration models all increased with restoration years in the topsoil, whereas the soil TP concentration remained relatively stable. The soil C:N and C:P ratios increased with increasing restoration years in the topsoil, whereas the N: P ratio was relatively stable over time. After ten years of reforestation, the SOC and TN concentrations decreased as the soil layer increased. The effects of the restoration model on the C: N ratios were greater in shallow soils. Our results suggest a complex reaction of SOC, soil TN, and soil TP concentrations and stoichiometry to the vegetation restoration mode, particularly in the topsoil. This research further improves the understanding of SOC, N, and P interactions and restricted nutrition, and provides relevant theoretical support for vegetation restoration in the southwest karst region.

**Keywords:** soil stoichiometry; soil nutrient; vegetation restoration; karst ecosystems

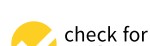



## 1. Introduction

As important components of soil nutrients, soil C, N, and P play a key ecological part in forest ecosystems. Their stoichiometry can effectively predict the soil nutrient saturation state [1], acting as a driver in the ecosystem material cycle, element balance, and biological survival. Since the Redfield ratio was proposed, stoichiometric ratios have received increasing attention worldwide for further determination of the limiting elements, nutrient cycling, and nutrient accumulation and balance in different terrestrial ecosystems [2,3]. They can promote understanding of how ecological processes respond to global change [4]. Considerable research has been undertaken in surveying soil C:N:P stoichiometry ratio patterns at the community, landscape, regional, continental, and global scales [5–9]. Land use

changes the vegetation biomass stock and the aboveground plant communities, affecting the biogeochemical cycles [10,11].

The "conversion of cropland to forest program" (CCFP) initiated by China has promoted soil nutrient cycling and maintained soil quality by changing the vegetation community structure and composition, and the quantity and quality of litter. Improvements in the soil quality affect plant production and ecosystem functioning [12]. Extensive studies have indicated that the concentrations and stoichiometry of the soil organic carbon, soil total nitrogen, and the soil total phosphorus vary with vegetation type [13,14]. They also vary considerably in different soil layers owing to varied nutrient inputs and outputs among the different soil horizons [12,15,16]. Zhang et al. [13] documented that vegetation in the process of restoration is conducive to the accumulation of soil C, N, and P. Gao et al. [14] observed that C and N concentrations and stock in subtropical orchard and farmland soils were higher than those in forest soils. Cao and Chen [17] reported that natural forests have a greater carbon storage capacity than plantations. Xu et al. [12] found that the overall C:N, C:P, and N:P increased with recovery year on the Loess Plateau. Therefore, it is of great significance to study the variation of soil C, N, and P concentrations and their stoichiometric ratios in soil under different vegetation restoration methods.

The karst area in southwest China is one of the three largest-scale karst concentrated distribution regions in the world. It is mainly distributed in southwest China, centered around Guizhou, covering of about $5.5 \times 10^5$ km$^2$ [18]. Within the last century, a large proportion of the karst region has severely deteriorated due to intensive human activities [19]. Since the 1970s, China has carried out ecological function restoration and protection projects across the country, aiming to restore and improve our country's ecological environment. Among them, the Grain to Green project is a key item in many initiatives [20–22]. After the implementation of the project, the local ecological environment has been significantly improved, and the biodiversity in the forest has also greatly increased. There have been a large number of relevant studies on the ecological stoichiometry of carbon, nitrogen, and phosphorus in karst areas [23–25]. Yu et al. [24] used the space-for-time substitution method to study the soil stoichiometric characteristics of different vegetation types. Some studies have shown the response of the topsoil and the microbial C: N: P to vegetation succession [23]. Experiments on the stoichiometric characteristics of the vegetation and the litter layers have been extensively studied [25]. However, there are few reports on the study of soil nutrient stoichiometry from the perspective of long-term site monitoring and monitoring at different soil depths.

In our study, we selected the soil from depressions between karst hills in southwest China as research objects to study temporal and vertical change in the soil SOC, TN, and TP under seven different restoration models. The main purposes of this study were to investigate the following: (1) How did the SOC, soil TN, soil TP, and their stoichiometric ratios change under different models with restoration years? (2) How did the SOC, soil TN, soil TP, and their stoichiometric ratios change under different models across different soil layers?

The findings of our study contribute to the exploration of maintenance mechanisms in forest ecosystems and support forest management. These results will provide a scientific basis to better understand variation in the soil quality after karst land use conversion and enhancing ecosystem services.

## 2. Materials and Methods

### 2.1. Study Area

This study was conducted in Guzhou Village (24°53′–24°55′ N, 107°56′–107°58′ E), distributed in the karst area of Maonan Autonomous Region, Huanjiang County, northwest Guangxi Zhuang Autonomous Region, Southwest China. This region has a subtropical monsoon climate with an annual average temperature of 16.5–20.5 °C, an annual average rainfall of 1389 mm, and an annual average cumulative sunshine duration of 1451 h. The wet season usually starts in April and lasts until September, accounting for approximately

70% of the annual rainfall. The annual average frost-free period lasts for 290 days [26]. The region is a typical karst landform with gentle valleys and steep mountains. The soils are calcareous lithosol (limestone soil) [27]. Due to the fragility of the ecological environment and the impact of human activity, the vegetation types in this area are diverse, forming a pattern of different forest ecosystems such as shrubs, arbor shrubs, artificial forests, and arbor bushes [28,29]. In the early 21st century, with the support of the "conversion of cropland to forest program" (CCFP), the degraded sloping farmland was mostly abandoned and restored under different restoration strategies.

### 2.2. Soil Sampling

We selected seven CCFP locations on the slopes to represent the different restoration models. Four restoration models were established using one of the following tree species: *Toona sinensis* (XC), *Castanea mollissima* (BL), *Zenia insignis* (RD), *and Citrus reticulate* (GJ). One restoration model used the *Zenia insignis* and Guimu-1 elephant grass (GR), and the other used Guimu-1 elephant grass (GM). The remaining restoration model was abandoned farmland (LH) with natural regeneration of vegetation. We selected a 20 × 20 m plot at each site in December 2007, and the sites were similar in slope and aspect. Each restoration model selects 3 sampling points as repetitions [25]. The first survey was conducted in 2007 to collect the topsoil. The survey was repeated in 2017 with soil samples from different layers (0–10, 10–20, 20–30, 30–50, and 50–100 cm). After removing the leaf litter and humus layer from the soil surface, the soil layers below were sampled in each plot using a shovel. Soil samples from the same layer were mixed in one sampling bag for each sampling site. A total of 105 soil samples were collected, including 7 restoration models, 3 sites, and 5 soil layers. All the samples were air-dried, the roots and stones were removed, and the samples were passed through 0.15 mm sieves for the SOC, TN, and TP analyses [25].

### 2.3. Chemical Analysis

The SOC concentration was determined by potassium dichromate ($K_2Cr_2O_7$) oxidation by heating in an oil bath. The total N (TN) concentration was analyzed using an automatic Kjeldahl nitrogen analyzer. Total Kjeldahl nitrogen is the sum of the organic bounded nitrogen groups and the ammonium-nitrogen. The total P (TP) concentration was determined using $HClO_4$–$H_2SO_4$ digestion followed by a Mo–Sb colorimetric assay [30].

### 2.4. Data Analysis

The single-sample Kolmogorov–Smirnov test was used to test the normal distribution of soil C, N, and P and the stoichiometric ratio, and the Pearson correlation analysis method was used to determine the relationship between soil C, N, and P and their proportions. One-way analysis of variance (ANOVA) with a least significant analysis (LSD) post hoc test of significance was used to compare the soil C, N, and P and their ratios among the seven restoration models, restoration years, and five soil layers. Two-way ANOVAs were used to test the effects of restoration models, restoration time, soil depth, and their interaction with soil C, N, and P concentrations and stoichiometry. The Kruskal–Wallis test, which is a non-parametric test, was used to test the significance when the sample data showed unequal variance. All the statistical analyses and graphics were performed using the R software platform 4.1.2 (R Core Team, 2021, R Foundation for Statistical Computing, Vienna, Austria. URL https://www.R-project.org, accessed on 30 January 2022). The statistical significance was set at $p < 0.05$.

## 3. Results

### 3.1. Temporal Changes in the SOC, TN, and TP Concentrations and Ecological Stoichiometry under Different Models

The restoration model and time were found to have significant effects on the SOC, TN, and TP, and their stoichiometry, except for the effect of time on the TP. The effects of the interaction were found to be significant, except for the effect on the TP and the C:N

(Table 1). In the GJ model, the SOC, TN, TP, C:P, and N:P decreased. The TN and N:P in the XC model and the N:P ratio in the BL model decreased, and the indices in the other models increased. However, only the TN in the LH model, the C:N in the XC and RD models, and the C:P and the N:P ratios in the GM model showed significant changes with restoration years (Figure 1).

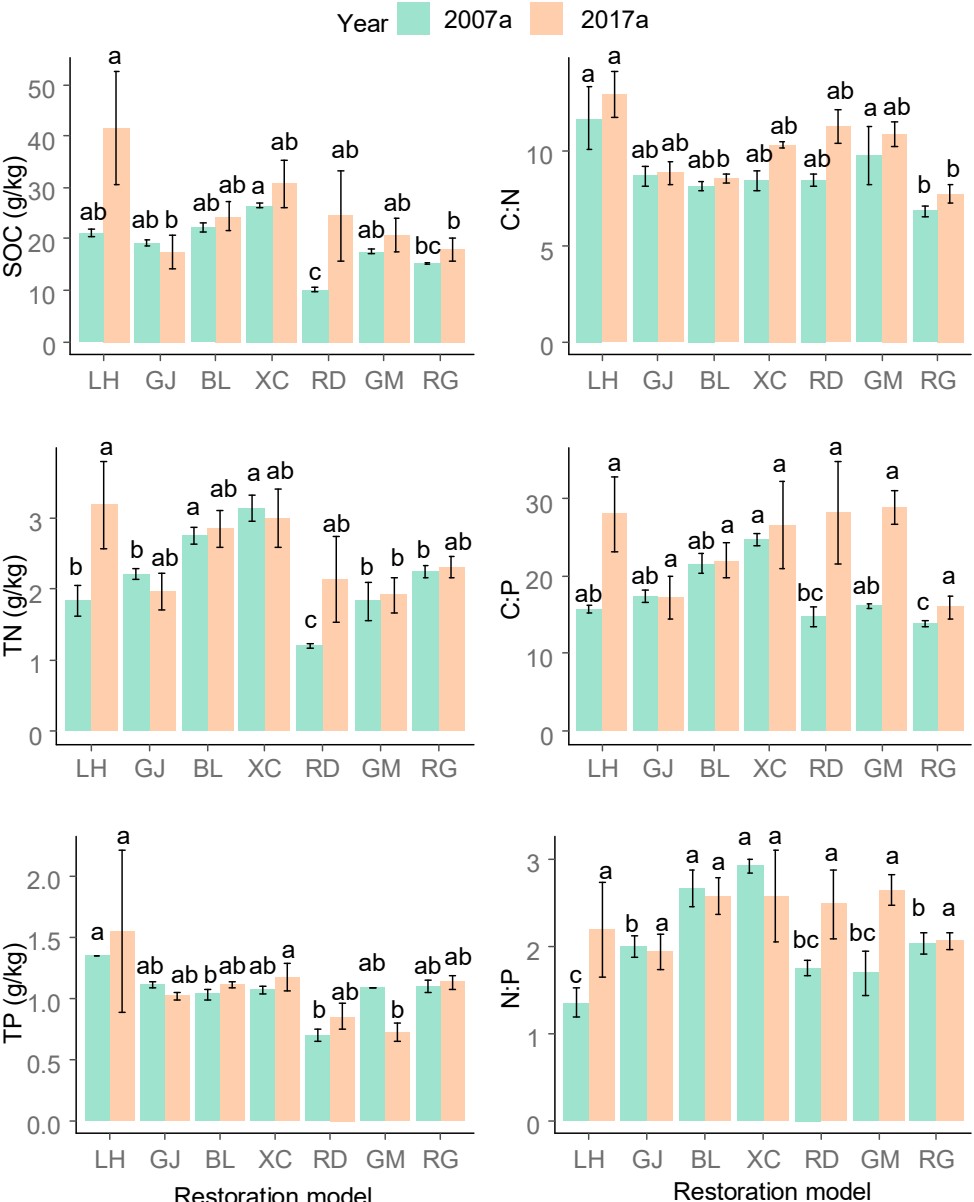

**Figure 1.** Contents of SOC, TN, and TP in the different restoration models in 2007a and 2017a. Different small letters mean significant differences in different restoration models. **Note:** 2007a and 2017a refer to in the 2007 year and in the 2017 year. *Toona sinensis* (XC), *Castanea mollissima* (BL), *Zenia insignis* (RD), *Citrus reticulate* (GJ), the *Zenia insignis* and Guimu-1 elephant grass (GR), and the other Guimu-1 elephant grass (GM). The remaining restoration model was abandoned farmland with natural regeneration of vegetation (LH). The same below.

In the early stage, the SOC and TN in the XC and BL models were higher than those in the other models, and the lowest values for the SOC, TN, and TP were observed in the RD model. At the later recovery stage, compared with the other restoration models, the SOC, TN, and TP in the LH model were significantly higher. The SOC in the GJ model and the TN and TP in the GM model showed the lowest values. The C:N ratio in the LH model

showed the highest value, and the RG model showed the lowest value twice. In the early stages, the C:P and N:P ratios were highest in the XC model, the lowest value of C:P was in the RG model, and the N:P value was the lowest in the LH model. The C:P and N:P ratios showed no significant changes in the different models at the later recovery stage (Figure 1).

**Table 1.** F and *p*-values for the independent factors (models, year) and their interactions.

|  | SOC | | TN | | TP | | C:N | | C:P | | N:P | |
| --- | --- | --- | --- | --- | --- | --- | --- | --- | --- | --- | --- | --- |
|  | **F** | **Sig.** | **F** | **Sig.** | **F** | **Sig.** | **F** | **Sig.** | **F** | **Sig.** | **F** | **Sig.** |
| Models | 11.25 | <0.01 | 15.91 | <0.01 | 7.74 | <0.01 | 24.32 | <0.01 | 8.59 | <0.01 | 9.99 | <0.01 |
| Year | 24.36 | <0.01 | 10.57 | <0.01 | 0.07 | NS | 24.94 | <0.01 | 43.83 | <0.01 | 12.26 | <0.01 |
| Models × Year | 5.07 | <0.01 | 5.75 | <0.01 | 1.64 | NS | 1.9 | NS | 6.81 | <0.01 | 5.75 | <0.01 |

*3.2. Vertical Changes in the SOC, TN, and TP Concentrations and Ecological Stoichiometry under Different Models*

The effects of different models on the SOC, TN, and TP concentrations, and their stoichiometric ratios, varied in different soil layers at the later recovery stage (Figures 2 and 3). The restoration model and the soil layer had significant effects on these indices, except for the effect of the soil layer on the TP, and the interaction effects were not significant (Table 2).

The soil nutrient concentrations and their stoichiometric ratios in different soil layers were changed by the different restoration models. The SOC at the 0–10 cm depth in the LH model was higher than that of the GJ and RG models, whereas the differences were not significant in the other models. The SOC content decreased with soil depth, but the difference in SOC content between soil layers in the same model was not significant. The soil TN content at the 0–10 cm depth was significantly higher in the LH model than in the GM model, and the soil TN content in the BL model was the highest of the soil layers. Although the soil TN content at the 30–50 cm depth in the BL model was higher than that of the LH, RD, and GM models, it did not significantly differ among the other models. The SOC and the TN showed similar vertical distributions. The soil TP content at the 0–10 cm depth in the LH and XC models was significantly higher than that in the GM model, and at the 20–30 cm depth in the RG model it was significantly higher than that in the GM model, whereas no significant differences were found among the other models. Compared with the soil C and N, the tendency of the soil TP to decrease with soil depth was less pronounced and relatively stable overall (Figure 2).

The average soil C:N, C:P, and N:P ratios in the study area were 10.07, 23.78, and 2.35, respectively. The effect of restoration model on the C:N ratios at the 0–30 cm depth was significant, whereas the effects in the subsoil were highly limited. The soil C:N ratios of each soil layer in the LH model were the highest. The soil C:N ratios were significantly higher in the RD model than in the GJ, BL, and RG models at the 0–10 cm depth and in the RG models at the 10–20 cm depth. The soil C:N ratios at the 20–30 cm depth in the LH model were significantly higher than those in the RG model (Figure 3). The topsoil C:P ratio of the GM model was the highest, which was significantly higher than that of the RG model. At a depth of 30–50 cm, the soil C:P ratio in the BL model was significantly higher than that of the GM and RG models, and did not significantly differ among other soil layers and models. The highest N:P ratios were found in the BL model among all the soil layers, and there were no significant differences in the N:P ratios at the 0–10 cm and 10–20 cm depths among all the models. At the 20–30 cm depth, the N:P ratios in the BL model were significantly higher than in the RD and RG models. At the 30–50 cm depth, the N:P ratios in the BL model were the highest, except for the XC model. N:P ratios at the 50–100 cm depth in the BL model were significantly higher than in the LH and RG models (Figure 3).

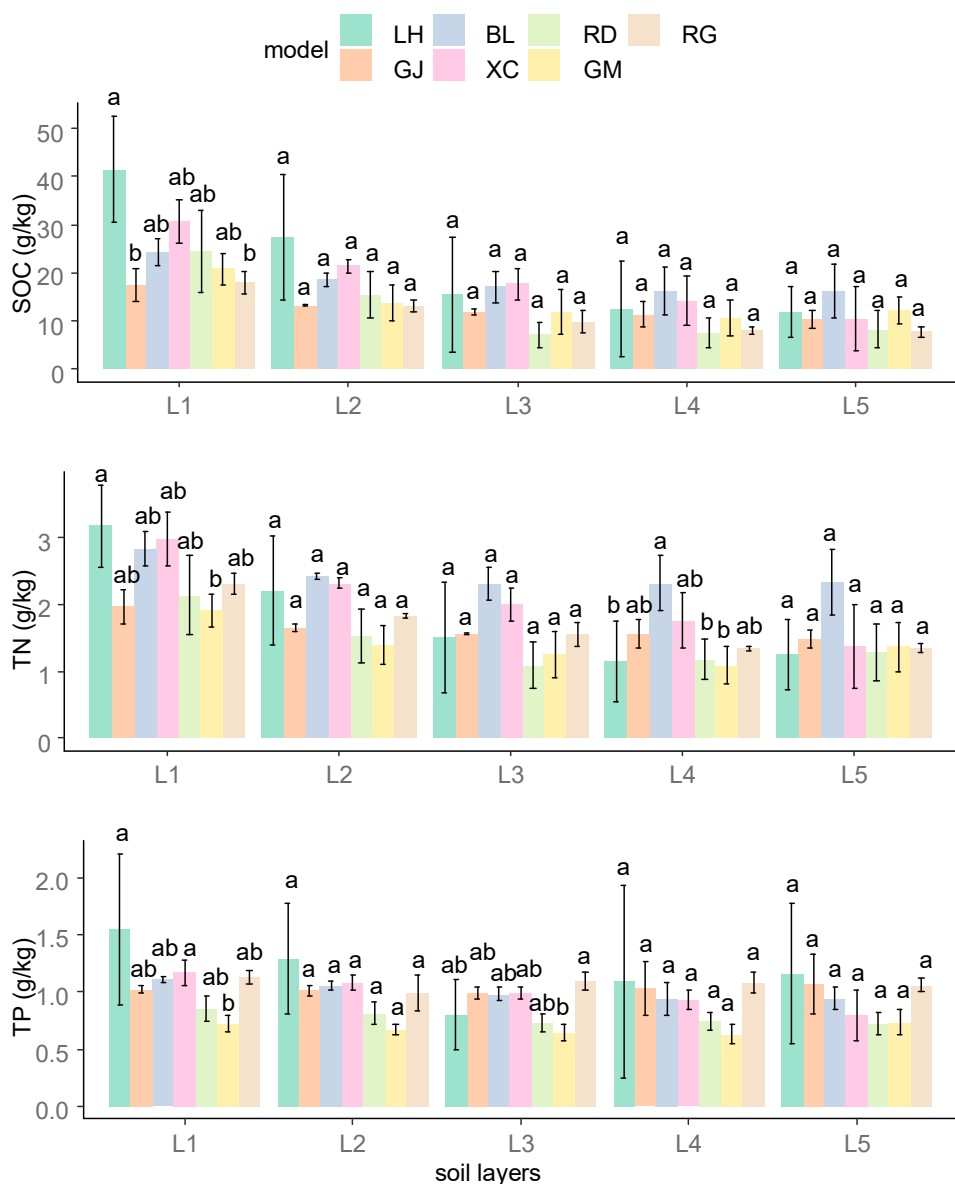

**Figure 2.** Contents of SOC, TN, and TP in the different restoration models for each soil layer. Different small letters mean significant differences among restoration models in the different soil layers.

**Table 2.** F and *p*-values of the independent factors (models, layer) and their interactions.

|  | SOC | | TN | | TP | | C:N | | C:P | | N:P | |
|---|---|---|---|---|---|---|---|---|---|---|---|---|
|  | **F** | **Sig.** | **F** | **Sig.** | **F** | **Sig.** | **F** | **Sig.** | **F** | **Sig.** | **F** | **Sig.** |
| Models | 8.54 | <0.01 | 13.79 | <0.01 | 7.2 | <0.01 | 25.73 | <0.01 | 13.44 | <0.01 | 16.9 | <0.01 |
| Layer | 27.01 | <0.01 | 24.46 | <0.01 | 1.87 | NS | 22.86 | <0.01 | 36.5 | <0.01 | 19.62 | <0.01 |
| Models × Layer | 1.44 | NS | 1.28 | NS | 0.56 | NS | 1.27 | NS | 1.38 | NS | 1.13 | NS |

NS: not significant at *p* > 0.05.

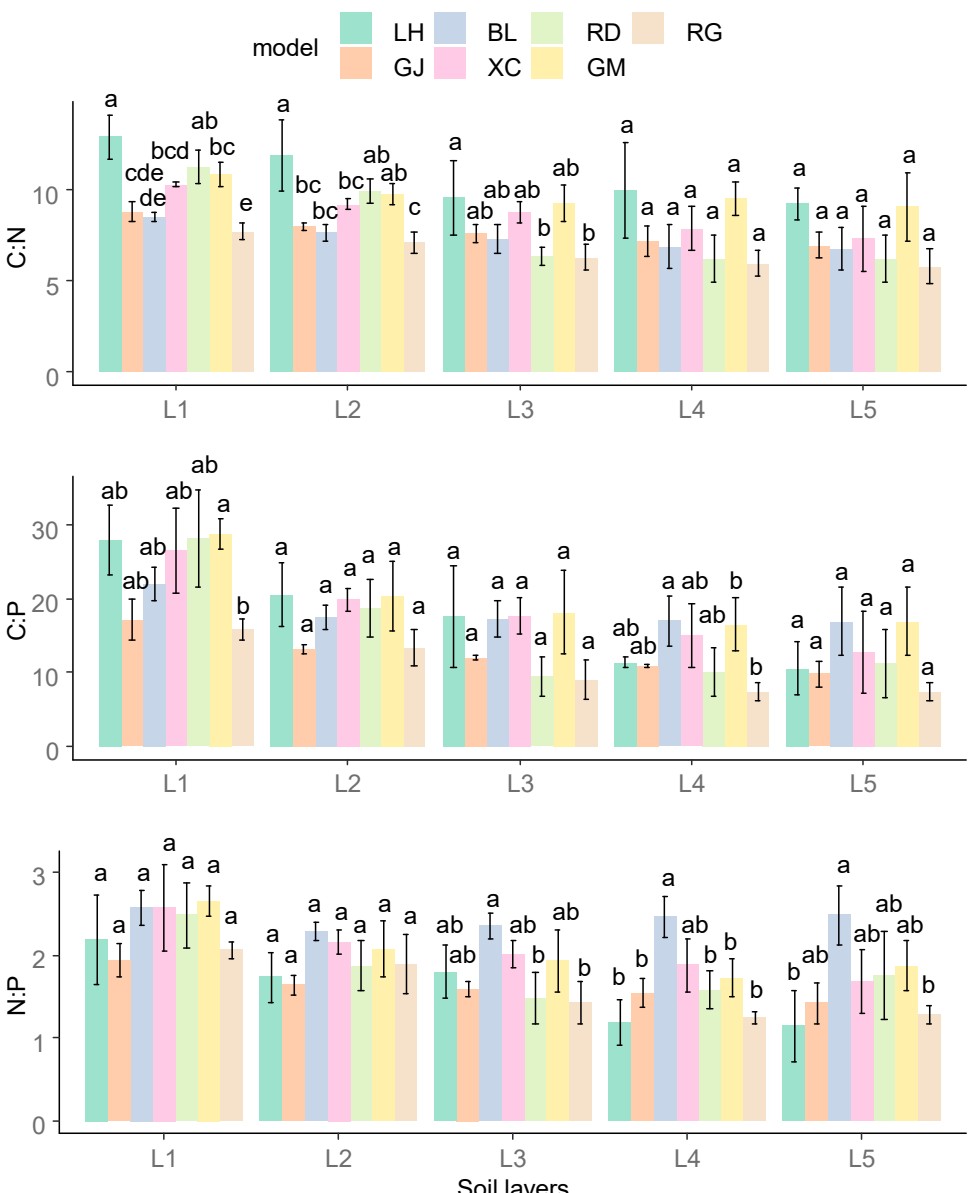

**Figure 3.** C:N, C:P, and N:P ratios in the different restoration models for each soil layer. Different small letters mean significant differences among restoration models in the different soil layers.

### 3.3. Relationships between SOC, TN, and TP, and Ecological Stoichiometry

As shown in Figure 4, in the early stages, the SOC had a significant positive correlation with soil TN and TP ($p < 0.01$), whereas no clear relationship was found between soil TN and TP ($p > 0.05$). The SOC and the TN were significantly positively correlated with the C:P and the N:P ratios ($p < 0.01$) but there was no clear relationship with the C:N ratio ($p > 0.05$). The TP had a significant positive correlation with the C:N ratio ($p < 0.01$) but there was no clear relationship with the C:P and N:P ratios ($p > 0.05$). There was a significant negative correlation between the N:P and the C:N ratios ($p < 0.05$) and a significant positive correlation between the C:P ratios ($p < 0.01$). However, the relationship between the C:N and the C:P ratios was not significant ($p > 0.05$).

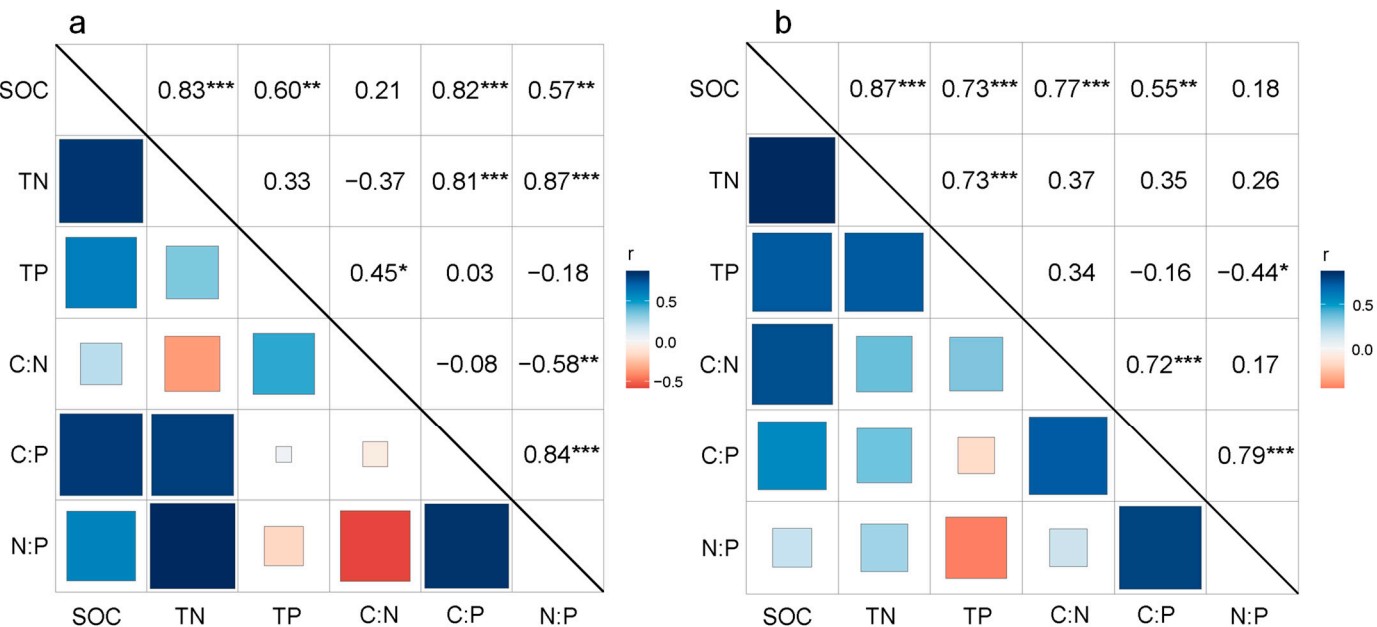

**Figure 4.** Correlation analysis between the SOC, TN, and TP, and Ecological Stoichiometry. **Note:** (**a**) in 2007 year, (**b**) in 2017 year. *, $p < 0.05$; **, $p < 0.01$; ***, $p < 0.001$.

At the later recovery stage, the SOC, TN, and TP were significantly positively correlated ($p < 0.01$). The SOC had a significant positive correlation with the C:N and the C:P ratios ($p < 0.05$) but there was no clear relationship with the N:P ratio ($p > 0.05$). The TN, TP, C:N, C:P, and N:P ratios were not significantly correlated ($p > 0.05$), with the exception that the TP was negatively correlated with the N:P ratio ($p < 0.05$). Positive correlations ($p < 0.05$) among the C:N, C:P, and N:P ratios were found, but there was no significant difference between the C:N and the N:P ratios (Figure 4).

## 4. Discussion

### 4.1. Responses of the SOC, TN, and TP Concentrations and Ecological Stoichiometry to Restoration Years and Restoration Models

Previous research has claimed that vegetation restoration can significantly improve the soil quality [18,25]. In this study, the SOC at 0–10 cm soil depth in different restoration models increased, except for in the GJ model in terms of restoration years. SOC is not only an important part of the soil organic carbon pool, but also the most important indicator of soil quality, which can reflect the soil fertility level and regional ecosystem evolutionary pattern [3,31]. The sources of SOC mainly include plants (aboveground litter, roots, and exudates) and microphyte substances (residues, metabolites, and extracellular secretions) [32,33]. During restoration years, a great deal of litter and organic matter input may be the reason for SOC accumulation [34]. As the recovery age increases, soil microbial and enzymatic activities are strengthened [35], thereby promoting soil nutrient transformation and storage. However, in the GJ model, the SOC decreased. This may be due to most of the nutrients in the GJ model being concentrated in the fruit, which are harvested artificially; the phenomenon of nutrient migration plays a prominent role. *Citrus reticulate* is evergreen and produces little litter, whereas other trees (*Zenia insignis*, *Toona sinensis*, *Castanea mollissima*) are deciduous. Our results also showed that the TN concentrations of 0–10 cm soil depth in different restoration models all increased, except for in the GJ and XC models, with restoration years. This indicates that planting *Toona sinensis* and *Citrus reticulate* is adverse to the accumulation of TN. For *Citrus reticulate*, because the N content was significantly affected by SOC, TN concentrations decreased with decreasing SOC concentrations. Plants absorb soil nitrogen during growth and then feed it back into the soil through litter. Therefore, the nitrogen absorption rate is lower than the rate of

release, causing an increase in soil TN [36]. In line with the previous research results [12,16], the change in the soil TP is not significant with the restoration year. The soil P is mainly formed by rock weathering [3] and is affected by the soil parent material, land use, and soil biogeochemical processes [37–39]. Due to the similar parent material and climate among the recovery models, the variability of TP was not as obvious as that of the SOC and soil TN, and the effect of restoration year on TP is not significant [12].

The C:N ratio reflects soil quality, affects the cycle of organic C and N in the soil, and determines the quality and accumulation rate of SOC [3,40]. Similar to the results of Xu et al. [12], our results showed that the C:N ratio increased in the surface soil layer with the restoration year. An important factor affecting soil C:N ratio is the change in SOC and the TN content [41]. The increase in the SOC concentrations was greater than the increase in soil TN concentrations and consequently, the C:N ratio increased. The C:P ratio is not only an indicator to reflect the availability of soil TP, but also an indicator to measure the immobilization of soil phosphorus by microorganisms in the soil [24,40]. In our study, we found that the soil C:P ratio increased with the restoration year. The SOC concentration in the soil increased, whereas that of the TP remained stable, causing an increase in the C:P ratio. However, in the GJ model, the SOC decreased with the restoration year, resulting in a decrease in the soil C:P ratio. Soil N:P as a predictor of nitrogen saturation not only reflects the availability of soil nutrients elements during plant growth, but is also used to evaluate the threshold of nutrient limitation [42–44]. Related research shows soil N:P ratio has a negative correlation with the plant growth rate [45], suggesting that the N:P ratio increases at low plant growth rates and decreases at high growth rates. In the present study, we found that the N:P ratio was reduced in the GJ, BL, and XC models over time. This may be due to the beneficial effects of these three restoration modes on soil improvement, resulting in a high plant growth rate and a decrease in the N:P ratio. However, the N:P ratio was increased in the LH, RD, GM, and RG models. With vegetation restoration, the contents of soil TN in these four models increased, whereas the TP content showed little variation, causing an increase in the soil N:P ratio. The N:P ratio had little change and remained relatively stable over time.

### 4.2. Responses of SOC, TN, and TP Concentrations and Stoichiometry to Soil Depth

Soil depth is a vital factor that determines spatial variation in the SOC, soil TN, and soil TP [46,47]. The present study implied that the SOC and the soil TN in the study area tended to decrease as the soil layer increased, which is in line with most previous studies' results [12,16,17,48,49]. Litter decomposition is an important source of soil nutrients, and this process occurs mainly in the topsoil [14], which increases the nutrient accumulation in the topsoil. As soil depth increases, organic matter input is limited due to a decrease in microbial decomposition activity and the root absorption [32,47]. The distribution pattern of SOC and TN decreased gradually from the surface layer to the deep layer. The SOC and soil TN are affected by soil parent material, litter decomposition, and plant uptake and utilization [50], resulting in considerable spatial variation. The soil layer in the karst area is shallower than in other areas, and the phenomenon of soil "surface accumulation" is more pronounced. However, the tendency of soil TP to decrease with soil depth was less pronounced and relatively stable in the present study, because soil TP is a sedimentary mineral with poor migration in soil, which is mainly affected by soil parent material [51–53]. Therefore, the variation of soil TP concentration with soil layer was relatively small.

Nutrient supply amounts and their coordination affect organism growth, population structure, species succession, and the ecosystem stability [54]. The average C:N ratio at a depth of 0–10 cm in this study was 10.07, which was lower than the mean for China (12.30) and the global mean (12.40) [40,55]. Soil C:N ratio is considered as a predictor of forest organic matter decomposition, and a low soil C:N ratio often indicates rapid decomposition of organic matter in soil and relative abundance of soil nitrogen [1,56]. This indicates that soil C accumulation and soil organic matter decomposition in the karst peak cluster depression are faster than average. This difference may be due to the distinctive geological

conditions of the karst area and the limestone soils. Research shows that calcareous soils which contain more $Ca^{2+}$ and $Mg^{2+}$ accumulate more soil carbon than non-calcareous soils, regardless of whether the calcium is natural or anthropogenic [57,58]. In the present study, we found that the restoration model had a significant effect on the C:N ratios at the 0–30 cm depth, which is inconsistent with the conclusion that the soil C:N ratios are insensitive to land-use change [16]. However, the effects of the restoration model on the C:N ratios at the 30–50 cm and the 50–100 cm depths are highly limited. This may be because deeper soil did not directly provide nutrients needed for plant growth and was relatively less mineralized. Consequently, the SOC and the soil TN concentrations' change rates gradually decreased with soil depth. Our study demonstrated that the soil C:N ratios among all the soil layers in the LH model were the highest, indicating that the soil material, energy cycle, and self-regulation were the best under the condition of minimal human disturbance.

A low C:P ratio increases the available P in the soil by promoting microbial decomposition of organic matter. The average soil C:P ratio in the 0–10 cm soil layer in our study area was 23.78, which was much lower than the mean for Chinese (52.70) and mean for global forest soils (81.90) [40,55]. This value indicates that the karst soils in South China are rich in P in the topsoil, which is consistent with previous research results. SOC was significantly positively correlated with TN, and the soil C:P and N:P showed similar trends under different restoration modes. The average soil N:P ratio in the 0–10 cm soil layer in our study area was 2.35, which was lower than the mean for Chinese (9.3) and mean for global forest soils (13.1) [59], indicating that the soil in the study area has sufficient levels of P, but is lacking in N during the early stages of restoration. Therefore, N element management should be strengthened during the recovery period.

## 5. Conclusions

We investigated changes in the soil nutrient contents and stoichiometry at different depths of 0–100 cm in seven plantation sites in a typical karst ecosystem (CCFP) in southwest China. The results showed that in addition to the effects of time and soil layer on TP, different afforestation methods, time and soil layers had significant effects on the SOC, TN, and TP and their stoichiometry. The SOC (except in the GJ model), TN (except in the GJ and XC models), and C:N and C:P ratios in different restoration models all increased with restoration years, whereas the soil TP concentration and N:P ratio remained relatively stable. The SOC and the TN concentrations had a decreasing trend as the soil layer increased, exhibiting the phenomenon of soil "surface accumulation". We found that the restoration model had a significant impact on the C:N ratios at the 0–30 cm depth, whereas the effects in the subsoil were limited. In the early stage of the Grain for Green project returning farmland to forests, N was lacking in the soil, and N application can promote plant growth. The results of this study further improve understanding of C, N, and P interactions and nutrient limitations, and will supply relevant theoretical support for vegetation restoration in the southwest karst region.

**Author Contributions:** Conceptualization, M.L. and H.D.; Funding acquisition, H.D.; Field sampling, H.D., K.L., L.Z., W.P. and T.S.; Methodology, F.Z. and T.S.; Writing—original draft, M.L. and H.D. All authors have read and agreed to the published version of the manuscript.

**Funding:** National Natural Science Foundation of China (42071073, 31971487, 32071846), Youth Innovation Promotion Association of the Chinese Academy of Sciences (2021366), Guangxi Key Research and Development Program (AB17129009), Hechi Distinguished Expert Program to Fuping Zeng, and Guangxi Bagui Scholarship Program to Dejun Li.

**Institutional Review Board Statement:** Not applicable.

**Informed Consent Statement:** The authors declare that the research was conducted in the absence of any commercial or financial relationships that could be construed as a potential conflict of interest.

**Data Availability Statement:** Not applicable.

**Conflicts of Interest:** The authors declare no conflict of interest.

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
