# Peer review of "Stoichiometric Variation in Soil Carbon, Nitrogen, and Phosphorus Following Cropland Conversion to Forest in Southwest China"

_forests, doi:10.3390/f13081155_

Round 1

Reviewer 1 Report

The paper deals with the problem of changes in the content of organic matter (C,N,P) in areas with different types of land cover. They are characterized by various biological properties, and the plant cover has a decisive influence on the fertility of these soils.
There are a few inaccuracies in the work:
"In December 2007, we sampled one 20 m × 20 m plot at each site, and the plots had a similar slope position, degree, and aspect" - is duplicated (line 109-112). Based on the result the year in line 111 should be 2017.
"sample was passed through 0.15-mm and 1-mm mesh sieves" is this correct?
What does "a", "b" "ab" and "bc" mean?
Why OX on fig 3 is described as "soil layers"? Shouldn't it be "restoration model"?

Author Response

Point 1: "In December 2007, we sampled one 20 m × 20 m plot at each site, and the plots had a similar slope position, degree, and aspect" - is duplicated (line 109-112). Based on the result the year in line 111 should be 2017.

Response 1: Thanks very much for your suggestion, we have deleted the duplicated line.

Point 2: "Sample was passed through 0.15-mm and 1-mm mesh sieves" is this correct?

Response 2: Yes, Thanks very much for your carefully review. We have cited relevant literature.

Point 3: What does "a", "b" "ab" and "bc" mean?

Response 3: Different small letters mean significant differences among restoration models in the different soil layers.

Reviewer 2 Report

The article raises a very important issue - especially from the point of view of the possibility of increasing the resources of organic carbon in anthropogenically degraded soils. The authors presented the results of many years of research on several research sites different in terms of species. The research was very well planned and carried out. The results are beyond doubt. The article should be published with minor changes.

1. How did the authors collect samples up to 100 cm deep for soils designated as lithosols? These soils are rocky and have no such deep solum.

2. The authors should specify the soil unit covered by the research. It is best to classify the soils according to the WRB or Soil Taxonomy.

3. Studies from China dominate the literature. The authors should extend the spectrum of the cited literature to include research from other regions.

4. You need to correct your spelling. With ratios, e.g. C: N, no space after the colon is needed (not - C: N)

5. On line 111 there is an unnecessary repetition of the sentence.

6. Line 116 - for what purpose has the A (humus) horizon been removed, since it is the most important genetic horizon in lithosols? Did the authors mean the organic horizon?

Some details are also in attached file.

Author Response

Point 1: How did the authors collect samples up to 100 cm deep for soils designated as lithosols? These soils are rocky and have no such deep solum.

Response 1: In this study, the returned farmland is mainly in the downslope and depression areas, and the soil is relatively thick on the upslope, most of which are over 50cm, and a few of which are up to 1m. Therefore, the land above 50cm is classified as 50-100 cm (L5)

Point 2: The authors should specify the soil unit covered by the research. It is best to classify the soils according to the WRB or Soil Taxonom.

Response 2: The soils The soils are calcareous lithosol (limestone soil). We have explained this in the manuscript. Please see in line 101.

Point 3: Studies from China dominate the literature. The authors should extend the spectrum of the cited literature to include research from other regions. 

Response 3: Yes, we have added. Thanks very much for your suggestion

Point 4:You need to correct your spelling. With ratios, e.g. C: N, no space after the colon is needed (not - C: N).

Response 4: Yes, thanks very much for your suggestion. We have revised.

Point 5:On line 111 there is an unnecessary repetition of the sentence.

Response 5: Yes, we have deleted.

Point 6:Line 116 - for what purpose has the A (humus) horizon been removed, since it is the most important genetic horizon in lithosols? Did the authors mean the organic horizon?

Response 6: Soil sampling to remove humus is mainly to take soil, rather than to take fallen leaves, which can effectively avoid the impact of fallen leaves on the soil. We mainly analyze the SOC, N, P in the soil, etc., if the fallen leaves are collected, it will have a great impact on the measurement results.

Reviewer 3 Report

The work is of great interest for studying the mechanism of restoration of forest soils.

I hope that the comments below will help improve your manuscript.

93 - the line " fall of 1,389 mm, and a mean annual cumulative sunshine duration of 1,451 h" needs to remove the comma

Figure 4. What do  mean   * * * ?

The paper presents few discussions of the data obtained on the content and intra-profile distribution of N, P, C and the ratio of these elements in soils and the significance of the studied indicators for the processes occurring in soils.

On the basis of what is the conclusion about the restoration of soils? It will be better to add the studied parameters for reference undisturbed soils for this territory.

Reducing the content of the studied N, P, C down the profile is standard for many soils and does it make sense to do such an analysis ? Maybe you had a hypothesis that you were testing?

To assess the recovery, it is better to consider not the content of the element, but their reserves in the studied layer.

In manuscript absent description of the depth of humus and sub-humus horizons before and after experiment, to judge the processes occurring inside the soil profile in different versions of the experiment and soil restoration.

Author Response

Point 1: 93 - the line " fall of 1,389 mm, and a mean annual cumulative sunshine duration of 1,451 h" needs to remove the comma

Response 1: Yes, We have remove the comma.

Point 2: Figure 4. What do  mean   * * * ?

Response 2: * * * means P < 0.001.

The paper presents few discussions of the data obtained on the content and intra-profile distribution of N, P, C and the ratio of these elements in soils and the significance of the studied indicators for the processes occurring in soils.

Point 3: On the basis of what is the conclusion about the restoration of soils? It will be better to add the studied parameters for reference undisturbed soils for this territory.

Response 3: Thanks for your valuable suggestions. The conclusion are that different afforestation methods, restoration time and soil layers had significant effects on the SOC, TN, TP and their stoichiometry. Abandoned land for natural regeneration of vegetation acts as a control.

Point 4: Reducing the content of the studied N, P, C down the profile is standard for many soils and does it make sense to do such an analysis ? Maybe you had a hypothesis that you were testing?

Response 4: Studies about the effects of vegetation restoration on soil nutrient concentrations and C:N:P stoichiometry are mainly focused on topsoil (0–20 cm). Recent studies have suggested that the nutrient status of deep soil may also be affected by long-term vegetation restoration. Therefore, studies about the effect of The purpose of this study was to understand how did the SOC, soil TN, soil TP, and their stoichiometric ratios change under different models across different soil layers?

Point 5: To assess the recovery, it is better to consider not the content of the element, but their reserves in the studied layer.

Response 5: Yes, Thanks for your valuable suggestions. In this study, changes of nutrient content and its stoichiometric ratio were considered. Soil nutrient content could also reflect soil fertility.

Point 6: In manuscript absent description of the depth of humus and sub-humus horizons before and after experiment, to judge the processes occurring inside the soil profile in different versions of the experiment and soil restoration.

Response 6: Thanks for your valuable suggestions. This study did not investigate the depth of humus and sub-humus horizons, which can be supplemented in the future.

Reviewer 4 Report

Dear authors,

I have some suggestions and some comments to your work.

Line 53: I suggest the following change: “dfferent” by “different”

 Line 53: I suggest the following change: “Zhang et al. (2019) found that vegetation restoration is conducive to the accumulation of soil C, N, and P [13]” by “Zhang et al. [13] found that vegetation restoration is conducive to the accumulation of soil C, N, and P”

The same suggestion for Gao et al. (2014) (line 54). Cao and Chen (2017) (line 56). Xu et al. (2018) (line 58).  Yu et al. (2014) (line 70). Xu et al. (2019) (line 263),

Line 76: I suggest changing the word “continuous” to another one or another expression.

For example, if you use sensors to record soil moisture you have “continuous monitoring” but in your case, I would not say that you have done continuous monitoring.

However, the sentence "there are few reports on the study of soil nutrient stoichiometry from the perspective of continuous site monitoring and monitoring at different soil depths" is a general sentence and it can be left as it is.

 Lines 109-112: Do these sentences mean the same: “We sampled one plot (20 × 20 m) in December 2007 at each site, and the plots had a similar slope position, degree, and aspect” “In December 2007, we sampled one 20 m × 20 m plot at each site, and the plots had a similar slope position, degree, and aspect”?

 Lines 119-121: Please, in the sentence “All the samples were air-dried, the roots and stones were removed, and the sample was passed through 0.15-mm and 1-mm mesh sieves for the SOC, TN, and the TP analyses”, can you clarify for which soil nutrient you used 0.15-mm and for which 1-mm in the analyses?

 Line 125: Considering that total nitrogen and total kjeldahl nitrogen are not the same, because the total kjeldahl nitrogen is the sum of the organic bounded nitrogen groups and the ammonium-nitrogen, you can write something like that:

“The total N (TN) concentration was analyzed using an automatic Kjeldahl nitrogen analyzer. Total Kjeldahl nitrogen is the sum of the organic bounded nitrogen groups and the ammonium-nitrogen”

Unless you used a variation of the Kjeldahl method that includes nitrates.

Line 127: Data analysis

1)      I wonder if you should have used Spearman correlation instead of Pearson correlation.

2)      You only mention “One-way analysis of variance (ANOVA)”, however, you use two-way anova in order to obtain the interaction of two independent factors [models, year].

 Line 158: What does it mean the “a” accompanying 2007 and 2017 in Figure 4 and in Figure 1. I suppose it means that you have considered independently the two years. Can you write this in the caption of the figure?

 Line 206: It is not clear the caption of figure 2: “Contents of SOC, TN, and TP in the different restoration models. Different small letters mean significant differences in the different soil layers”.

I suggest: Contents of SOC, TN, and TP in the different restoration models for each soil layer. Different small letters mean significant differences among restoration models in the different soil layers.

See if the caption of figure 3 also needs a change.

Line 231: I think Figure 4 is confusing. It seems that you have to fold the box diagonally to see the meaning of the interaction. But the authors have decided to do so and I have nothing to say.

 Line 237-238: “As an important part of SOC pool, the SOC..”

I suggest writing SOC in words. In this way, it becomes clearer that oxidizable organic carbon (SOC) is part of the soil organic carbon pool. “As an important part of soil organic carbon pool, the SOC”

Author Response

Point 1: Line 53: I suggest the following change: “dfferent” by “different”.

Response 1: Yes, We have revised.

Point 2: Line 53: I suggest the following change: “Zhang et al. (2019) found that vegetation restoration is conducive to the accumulation of soil C, N, and P [13]” by “Zhang et al. [13] found that vegetation restoration is conducive to the accumulation of soil C, N, and P”

The same suggestion for Gao et al. (2014) (line 54). Cao and Chen (2017) (line 56). Xu et al. (2018) (line 58).  Yu et al. (2014) (line 70). Xu et al. (2019) (line 263),.

Response 2: Thanks for your carefully review. We have revised.

Point 3: Line 76: I suggest changing the word ”continuous” to another one or another expression. For example, if you use sensors to record soil moisture you have “continuous monitoring” but in your case, I would not say that you have done continuous monitoring. However, the sentence "there are few reports on the study of soil nutrient stoichiometry from the perspective of continuous site monitoring and monitoring at different soil depths" is a general sentence and it can be left as it is.

Response 3: Thanks for your valuable suggestions. We have replaced continuous monitoring with long-term monitoring.

Point 4: Lines 109-112: Do these sentences mean the same: “We sampled one plot (20 × 20 m) in December 2007 at each site, and the plots had a similar slope position, degree, and aspect” “In December 2007, we sampled one 20 m × 20 m plot at each site, and the plots had a similar slope position, degree, and aspect”?

Response 4: Yes, we have deleted the duplicated line.

Point 5: Lines 119-121: Please, in the sentence “All the samples were air-dried, the roots and stones were removed, and the sample was passed through 0.15-mm and 1-mm mesh sieves for the SOC, TN, and the TP analyses”, can you clarify for which soil nutrient you used 0.15-mm and for which 1-mm in the analyses?

Response 5: The sample was passed through 0.15 mm sieves for the SOC, TN, and the TP analyses. We have cited relevant literature.

Point 6: Line 125: Considering that total nitrogen and total kjeldahl nitrogen are not the same, because the total kjeldahl nitrogen is the sum of the organic bounded nitrogen groups and the ammonium-nitrogen, you can write something like that:

“The total N (TN) concentration was analyzed using an automatic Kjeldahl nitrogen analyzer. Total Kjeldahl nitrogen is the sum of the organic bounded nitrogen groups and the ammonium-nitrogen”

Unless you used a variation of the Kjeldahl method that includes nitrates.

Response 6: Thanks for your valuable suggestions.

Point 7: Line 127: Data analysis

1) I wonder if you should have used Spearman correlation instead of Pearson correlation.

2) You only mention “One-way analysis of variance (ANOVA)”, however, you use two-way anova in order to obtain the interaction of two independent factors [models, year].

Response 7: 1) The data were linearly fitted before analysis, and the results showed that there was a linear relationship between variables, which met Pearson correlation requirements. 2) Thanks for your carefully review, we have added in the data analysis.

Point 8: Line 158: What does it mean the “a” accompanying 2007 and 2017 in Figure 4 and in Figure 1. I suppose it means that you have considered independently the two years. Can you write this in the caption of the figure?

Response 8: Yes, we have added in the note.

Point 9: Line 206: It is not clear the caption of figure 2: “Contents of SOC, TN, and TP in the different restoration models. Different small letters mean significant differences in the different soil layers”. I suggest: Contents of SOC, TN, and TP in the different restoration models for each soil layer. Different small letters mean significant differences among restoration models in the different soil layers. See if the caption of figure 3 also needs a change.

Response 9: Yes, thanks for your valuable suggestions. We have revised.

Point 10: Line 231: I think Figure 4 is confusing. It seems that you have to fold the box diagonally to see the meaning of the interaction. But the authors have decided to do so and I have nothing to say.

Response 10: Thanks for your carefully review, we have revised, please see Figure 4.

Point 11: Line 237-238: “As an important part of SOC pool, the SOC.”

I suggest writing SOC in words. In this way, it becomes clearer that oxidizable organic carbon (SOC) is part of the soil organic carbon pool. “As an important part of soil organic carbon pool, the SOC”

Response 11: Thanks for your valuable suggestions, we have revised.

Round 2

Reviewer 4 Report

The article has been improved by incorporating the suggestions and comments of the reviewers.

The article has minor text editing. For example:

Lines 124-126: The following sentence does not need quotation marks: “The total N (TN) concentration was analyzed using an automatic Kjeldahl nitrogen analyzer. Total Kjeldahl nitrogen is the sum of the organic  bounded nitrogen groups and the ammonium-nitrogen”.

Line 159: The following sentence "the N:P value is the lowest in the LH model" should be "the N:P value was the lowest in the LH model".

Line 225: There is an ectra "c" in "was no c clear".

Line 231: I wonder if the following sentence "it was no clear relationship with" should be  "there was no clear relationship with..."

Line 348: "microphytes Substances" should be "microphytes substances"

Line 282: The authors should remove "to used" in  "is also used to used to evaluate".

Author Response

Point 1: Lines 124-126: The following sentence does not need quotation marks: “The total N (TN) concentration was analyzed using an automatic Kjeldahl nitrogen analyzer. Total Kjeldahl nitrogen is the sum of the organic  bounded nitrogen groups and the ammonium-nitrogen”.

Response 1: Yes, we have revised.

Point 2: Line 159: The following sentence "the N:P value is the lowest in the LH model" should be "the N:P value was the lowest in the LH model".

Response 2: Yes, we have revised.

Point 3: Line 225: There is an ectra "c" in "was no c clear".

Response 3: Yes, we have deleted.

Point 4: Line 231: I wonder if the following sentence "it was no clear relationship with" should be  "there was no clear relationship with..."

Response 4: Yes, thanks for your carefully review. We have revised.

Point 5: Line 348: "microphytes Substances" should be "microphytes substances"

Response 5: Yes, we have revised.

Point 6: Line 282: The authors should remove "to used" in  "is also used to used to evaluate".

Response 6: Yes, we have revised.